# A Novel Network Framework on Simultaneous Road Segmentation and Vehicle Detection for UAV Aerial Traffic Images

**DOI:** 10.3390/s24113606

**Published:** 2024-06-03

**Authors:** Min Xiao, Wei Min, Congmao Yang, Yongchao Song

**Affiliations:** 1Project Construction Management Company of Jiangxi Transportation Investment Group Co., Ltd., Nanchang 330108, China; xiaom0649@gmail.com (M.X.); weimin2024@gmail.com (W.M.); 2School of Computer and Control Engineering, Yantai University, Yantai 264005, China; weareyoung@s.ytu.edu.cn; 3School of Computer and Artificial Intelligence, Zhengzhou University, Zhengzhou 450001, China

**Keywords:** UAV aerial traffic images, road segmentation, vehicle detection, attention mechanism, cascade feature fusion

## Abstract

Unmanned Aerial Vehicle (UAV) aerial sensors are an important means of collecting ground image data. Through the road segmentation and vehicle detection of drivable areas in UAV aerial images, they can be applied to monitoring roads, traffic flow detection, traffic management, etc. As well, they can be integrated with intelligent transportation systems to support the related work of transportation departments. Existing algorithms only realize a single task, while intelligent transportation requires the simultaneous processing of multiple tasks, which cannot meet complex practical needs. However, UAV aerial images have the characteristics of variable road scenes, a large number of small targets, and dense vehicles, which make it difficult to complete the tasks. In response to these issues, we propose to implement road segmentation and on-road vehicle detection tasks in the same framework for UAV aerial images, and we conduct experiments on a self-constructed dataset based on the DroneVehicle dataset. For road segmentation, we propose a new algorithm C-DeepLabV3+. The new algorithm introduces the coordinate attention (CA) module, which can obtain more accurate segmentation target location information and make the segmentation target edges more continuous. Also, the improved algorithm introduces the cascade feature fusion module to prevent the loss of detail information in road segmentation and to obtain better segmentation performance. For vehicle detection, we propose an improved algorithm S-YOLOv5 by adding a parameter-free lightweight attention module SimAM. Finally, the proposed road segmentation–vehicle detection framework is utilized to unite the C-DeepLabV3+ and S-YOLOv5 algorithms for the implementation of the serial tasks. The experimental results show that on the constructed ViDroneVehicle dataset, the C-DeepLabV3+ algorithm has an mPA value of 98.75% and an mIoU value of 97.53%, which can better segment the road area and solve the problem of occlusion. The mAP value of the S-YOLOv5 algorithm has an mAP value of 97.40%, which is more than YOLOv5’s 96.95%, which effectively reduces the vehicle omission and false detection rates. By comparison, the results of both algorithms are superior to multiple state-of-the-art methods. The overall framework proposed in this paper has superior performance and is capable of realizing high-quality and high-precision road segmentation and vehicle detection from UAV aerial images.

## 1. Introduction

In recent years, UAVs have developed at a rapid pace and have a wide range of applications. With the characteristics of strong environmental adaptability, flexible scheduling, and lower cost of use, UAVs have a wide range of applications in intelligent transport systems, fire rescue, and environmental monitoring [1]. At the same time, there is an increasingly urgent demand for intelligent analysis and the processing of aerial images captured by UAVs. Using UAVs to extract road information and perform vehicle detection tasks has become a hot research topic [2].

In terms of road segmentation, road information includes road type, shape, location and sign lines, etc. Real-time and efficient road information plays an important role in intelligent transport, urban planning, and emergency response. The drone can fly quickly and efficiently, thus realizing the coverage of large road areas and adapting to complex inspection environments. It provides high-frequency data updates to realize the real-time detection of road scenes. UAV aerial imagery is an important source of information for military reconnaissance, emergency rescue, geological exploration, and other fields [3]. Although road segmentation from aerial images has received considerable attention, the task remains challenging due to factors such as complex shooting angles and irregular road segments. Traditional methods for extracting roads from UAV aerial images are costly, time-consuming, and have low segmentation accuracy. Mnih et al. [4] were the first to apply deep learning to aerial road segmentation tasks. Since then, researchers have gradually proposed various deep learning based methods to extract road regions from aerial images. These methods show higher efficiency compared to traditional algorithms. However, in the road segmentation task, there are still problems such as unclear road boundary delineation, inaccurate delineation results, and insensitivity to occlusion phenomena.

Deep learning-based target detection methods are widely used in the fields of traffic target detection, fire detection [5], and plant disease identification [6]. In terms of vehicle detection, detecting vehicle targets on the road based on drone aerial images can promote the construction of urban intelligent traffic [7,8]. Intelligent transportation is the current hot topic in the field of urban transport, and it is also the future direction of development in the field of transport. Vehicle detection based on UAV aerial images can provide basic data for the construction of intelligent transport, such as traffic flow and congestion, so as to provide more detailed and comprehensive basic data support for the realization of intelligent transport. Despite the rapid development of target detection methods based on deep learning, UAV aerial images are limited by the shooting angle and shooting distance. These make vehicle detection face difficulties such as a large number of small targets, complex background conditions, inconspicuous features, and occlusion. These unfavourable factors lead to unsatisfactory results of representative algorithms such as Faster R-CNN [9] and YOLOv5 for vehicle detection in aerial images.

In order to solve the above challenges, this paper proposes the C-DeepLabV3+ algorithm for road segmentation based on the DeepLabV3+ and S-YOLOv5 algorithms for vehicle detection based on the YOLOv5 algorithm, and the two improved algorithms are united to form a road segmentation–vehicle detection framework. The two new algorithms achieve better results through a series of improvements and are more applicable to UAV aerial traffic images.

The contributions of this paper are as follows:A new road segmentation algorithm, C-DeepLabV3+, is proposed, which introduces a Coordinate Attention module to obtain more accurate segmented target location information and employs a cascaded feature fusion module in the decoder. The algorithm significantly improves the performance of road segmentation in UAV aerial traffic images.A new target detection algorithm, S-YOLOv5, is proposed, which is based on the YOLOv5 algorithm with the addition of a parameter-free attention module known as SimAM. This algorithm enhances the ability to capture and detect the contextual features of vehicle targets, which effectively improves the performance of vehicle detection.We propose a road segmentation–vehicle detection algorithm under the same framework for accomplishing the serial tasks of road segmentation and vehicle detection on the road under UAV aerial images. Specifically, the C-DeepLabV3+ algorithm is utilized to accurately extract the road region in the aerial images, and then the vehicle targets on the road regions are accurately detected based on the extracted images using S-YOLOv5 so that the two tasks are serially combined together with high quality.

The rest of this paper is organized as follows. Section 2 introduces related works about road segmentation, road segmentation from UAV aerial images, object detection, and object detection from UAV aerial images. Section 3 describes the proposed method in detail. The experimental results are presented in Section 4. Finally, conclusions are given in Section 5.

## 2. Related Work

### 2.1. Road Segmentation Methods

Road segmentation is achieved through image segmentation techniques and is an important research direction in the field of computer vision. Currently, there are two methods for road segmentation in images: traditional methods and deep learning-based methods. The difference between the two lies in the different levels of road image features used. The former makes use of the medium and shallow features of the image, while the latter makes use of the medium and deep features of the image in order to obtain higher accuracy.

The traditional methods can be divided into three methods: feature-level [10,11,12,13,14,15], object-level [16,17,18], and knowledge-level [19,20,21,22,23,24] methods. Feature-level methods mainly use the geometric features of road images and radial features. Unsalan et al. [13] extracted the initial edges of roads and then used the Binary Ballon algorithm and graph theory to extract roads. Object-level methods take the whole road image as a processing unit, obtain the segmented region using the image segmentation technique, and then classify the road and nonroad regions. Knowledge-level methods identify and extract roads by building road-related knowledge models. For example, Liu et al. [24] used the information of the selected sample road sections to construct a geometric knowledge base of rural roads, and based on the knowledge base, they extracted complete rural roads from remote sensing images.

Deep learning has greatly advanced the field of computer vision [25], and many methods have also been used in the study of road segmentation. The Convolution Neural Network (CNN) can provide better road segmentation results than traditional methods by embedding many high-level and multiscale information. For example, Li et al. [26] proposed combining deep learning and linear integral convolution for road network segmentation from high-resolution images. Tao et al. [27] proposed Seg-Road, a connected structure road segmentation network based on the transformer and CNN.

In recent years, there has been a gradual increase in the use of end-to-end [28,29,30,31] neural network models to extract road regions. This method can extract more semantic information by doing feature fusion and feature refinement of semantic segmentation models, thus improving the accuracy of the model. For example, Zhou et al. [31] constructed a semantic segmentation neural network D-LinkNet based on LinkNet [32] by increasing the sensory field to improve the road segmentation accuracy. In addition, the adversarial generative network GAN [33,34,35,36] was also introduced into the research of road segmentation. Zhang et al. [34] proposed a remote sensing image road segmentation method based on an improved generative adversarial neural network, which was a conditional GAN with L2 loss that requires only a small number of samples for training and a small amount of computation.

### 2.2. Road Segmentation Methods Based on UAV Aerial Images

The changing angles and complex background of UAV aerial images make it more challenging to perform road segmentation on UAV aerial images. Starting with the work of Mnih, deep learning based methods have been used to detect road regions in aerial images. Saito et al. [37] used the CNN to directly segment roads and buildings in aerial images. To improve the performance of the CNN, Alshehhi et al. [38] replaced the fully connected layer in the CNN using global pooling and combined the shallow features of the roads in the neighboring regions with the deeper features of the convolutional neural network. Panboonyuen et al. [39] proposed an improved DCNN with Landscape Metrics to further reduce the number of misclassified road objects and finally optimized the road segmentation results using the Conditional Random Field (CRF). In order to improve the recall of the segmentation, Im et al. [40] proposed to improve the performance of CNN-based road segmentation algorithms using road label thickening and thinning.

Compared to the CNN, Fully Convolutional Networks (FCNs) are better for road segmentation because their extracted features contain more high-level abstract semantic information. Zhong et al. [41] applied the FCN to the task of extracting buildings and road regions from high spatial resolution images. To obtain better road classification results, Fu et al. [42] designed a multiscale network based on the FCN and used the CRF to refine the output maps. Based on U-Net [43], Zhang et al. [28] proposed ResU-Net for road recognition extraction, thus replacing the basic unit of U-Net with the residual unit to facilitate training and solve the degradation problem. Chen et al. [44] used multiple lightweight U-Nets, with the output of each U-Net network serving as the input of the next U-Net network. Meanwhile, they designed a multiobjective optimization structure to jointly train all the U-Nets, which solved the problem of the poor training effect of multiple independent models.

### 2.3. Object Detection Methods

Object detection is a hot issue in the current field of computer vision. The main task is to find the target position in the image and classify it. Object detection algorithms include two categories: traditional algorithms and deep learning-based algorithms.

Traditional object detection algorithms require hand-designed features, which can be divided into three steps: selecting the region, extracting the features, and classifying them. Dala et al. [45] proposed for the first time the HOG as a feature descriptor, which greatly improved the scale-invariant [46] and shape context-based [47] methods. The Deformable Part Model (DPM) [48] is the pinnacle of manually extracting features to achieve target detection based on manual labor. The development of deep learning has achieved a successful shift from manually designed features to feature extraction using convolutional neural networks. The CNN with different depth hierarchies allows for richer multilayered feature representations, thus resulting in significant improvements in object detection performance. Currently commonly used deep learning-based detection algorithms include two-stage and single-stage algorithms. Among them, the two-stage algorithms generate candidate regions first and then perform classification and regression for each candidate region. The single-stage algorithms perform both classification and regression. Girshirk et al. [49] combined region suggestion with the CNN and proposed a representative algorithm for two-stage object detection, R-CNN, which performed much better than traditional object detection algorithms. To address the shortcomings of the selection search algorithm, the Faster R-CNN algorithm utilizes the Region Proposal Network (RPN) to extract candidate frames, which improves the inference speed of the network. The accuracy values of the two-stage algorithms are higher than the single-stage algorithms, but the speeds are slower than the single-stage algorithms. The YOLO series algorithms [50,51,52,53,54] use regression to apply a single CNN to the whole image, thus dividing the image into grids of the same size to directly predict the category and bounding box of each grid. The SSD [55] algorithm combines the regression concept of YOLO with the anchor box mechanism of Faster R-CNN, which effectively handles objects of different sizes.

### 2.4. Object Detection Methods Based on UAV Aerial Images

Deep learning-based methods for object detection using UAVs have been widely used. For the problem of detecting congested regions in aerial images, Yang et al. [56] proposed the ClusDet detection network. It consists of four subnetworks: congested region extraction, iterative merging, scale estimation, and detection. The experimental results of this network on the VisDrone 2019 dataset [57] showed that ClusDet has higher detection accuracy compared to the baseline detection network, especially for small targets. In order to better run the target detection algorithm on UAV embedded devices, Zhang et al. [58] proposed the Slim YOLOv3 algorithm. This algorithm greatly reduced the number of parameters of the model by pruning the channels of the convolutional layer of YOLOv3, and it improved the detection accuracy and speed to some extent. Liu et al. [59] proposed an unanchored detector, Edge YOLO, for real-time detection on edge computing platforms. The algorithm used data enhancement methods to suppress overfitting during training, and it used a hybrid stochastic loss function to improve the detection accuracy of small targets. Zhu et al. [60] proposed TPH-YOLOv5 based on YOLOv5 by adding a prediction head to detect objects at different scales and replaced the original prediction head with Transformer Prediction Heads (TPHs) to explore the potential of prediction with a self-attention mechanism. They also integrated a CBAM [61] to find the attention region in scenes with dense targets, and they achieved good performance in target detection in UAV scenes.

## 3. Methods

### 3.1. Overview of the Road Segmentation–Vehicle Detection Framework

The two parts of the road segmentation and vehicle detection tasks are serial work and sequential nature, which together constitute the proposed road segmentation–vehicle detection framework. Therefore, it is crucial to improve the performance of the road segmentation and vehicle detection algorithms in order to ensure the performance of the segmentation–detection framework.

The implementation diagram of the road segmentation–vehicle detection framework consisting of the C-DeepLabV3+ and S-YOLOv5 algorithms is shown in Figure 1. Firstly, the C-DeepLabV3+ algorithm is used to segment the road region and nonroad region of the UAV aerial image, and only the road region is extracted from the original image to generate the corresponding road postsegmentation image. Then, the S-YOLOv5 algorithm is used to perform vehicle target detection on the road-extracted image. Since the detection image contains only the road region, only the vehicles on the road are detected, the types of vehicles detected are categorized into cars and trucks, and the corresponding detection image is generated. Eventually, the visualized image is obtained after the completion of the two tasks of road segmentation and on-road vehicle detection.

### 3.2. Road Segmentation Based on UAV Aerial Images

Road segmentation in UAV aerial images leads to inaccurate segmentation results due to problems such as variable road regions and occlusion. The existing methods make it difficult to achieve the ideal segmentation accuracy, especially in the segmentation of complex scenes. There are problems that the road edge is not smooth, and the nonroad and road regions cannot be accurately distinguished. In order to improve the precision and accuracy of road segmentation, a new C-DeepLabV3+ algorithm has been proposed. The C-DeepLabV3+ structure is shown in Figure 2. This road segmentation algorithm can handle ordinary roads and intersections.

On the one hand, this algorithm introduces the Coordinate Attention (CA) module, which pays more attention to the key characteristics of the road. In this way, more accurate segmentation target position information can be obtained so that the segmentation target edge is more continuous, and the accuracy of segmentation is effectively improved. On the other hand, a cascade feature fusion module is added to the decoder to fuse multiscale features at different levels. It can improve the network segmentation semantic information representation ability, reduce the loss of road information, and make up for the lack of traditional algorithms to extract roads on aerial images. The loss function used by the C-DeepLabV3+ algorithm is shown in Equation (Equation 1).
(1)$$DiceLoss=1-\frac{2\sum_{i=1}^{N}y_{i}{\hat{y}}_{i}}{\sum_{i=1}^{N}y_{i}+\sum_{i=1}^{N}{\hat{y}}_{i}}$$
where $y_{i}$ and ${\hat{y}}_{i}$ denote the labeled and predicted values of pixel *i*, respectively, and *N* is the total number of pixel points, which is equal to the number of pixels in a single image multiplied by the batch size.

#### 3.2.1. Coordinate Attention

The basic idea of the attention mechanism is to extract key information from the input features based on different weights. In DeepLabV3+, the small deep semantic feature map extracted by the backbone network leads to the abstraction of the obtained features, insufficient sensitivity to the important features, and a lack of sufficient spatial information. Our proposed C-DeepLabV3+ embeds feature location information into the channel attention by introducing the CA moudle. This improved algorithm not only captures channel information, but it also senses orientation and location information. It improves the sensitivity to important semantic information, enhances the attention to the segmented target, and helps the network model to segment the target object more accurately. The implementation process of the CA moudle is shown in Figure 3.

Firstly, pooling operation is performed on the input feature maps from the width and height directions to achieve global information extraction and retain the spatial structure information of the feature maps. When a feature map *X* of dimension *C × H × W* is input to CA moudle, the module will use two sets of pooling kernels (*H*,*1*) and (*1*,*W*) to average the pooling along the horizontal and vertical directions of each channel, respectively, to obtain two feature maps. The output with height *h* and width *w* on the cth channel are respectively shown in Equations (Equation 2) and (Equation 3).
(2)$$Z_{c}^{h}\left(h\right)=\frac{1}{W}\sum_{0⩽i⩽W}X_{c}\left(h,i\right)$$
(3)$$Z_{c}^{w}\left(w\right)=\frac{1}{H}\sum_{0⩽j⩽H}X_{c}\left(j,w\right)$$

In order to fully take advantage of the positional information and interchannel relationships captured in the previous stage, a transform operation is performed on the feature maps $Z_{c}^{h}$ and $Z_{c}^{w}$ using a shared 1 × 1 convolution. This allows the region of interest for each channel to be accurately captured, thus yielding an intermediate feature map *f* of size C/r×1×(H+W), as shown in Equation (Equation 4):(4)$$f=\delta \left(F_{1}\left(\left[Z^{h},Z^{w}\right]\right)\right)$$
where δ is a nonlinear activation function, *r* is the downsampling ratio, and *f* is an intermediate feature map that encodes spatial information in the horizontal and vertical directions.

Then *f* is sliced along the spatial dimension into two tensors $F_{h}$ and $F_{w}$ of sizes C/r×H×1 and C/r×1×W, respectively. $f_{h}$ and $f_{w}$ are transformed to the same channel size as the input *X* using two *1 × 1* convolutions $F_{h}$ and $F_{w}$. The transformation method is shown in Equations (Equation 5) and (Equation 6).
(5)$$g^{h}=\sigma \left(F_{h}\left(f^{h}\right)\right)$$
(6)$$g^{w}=\sigma \left(F_{w}\left(f^{w}\right)\right)$$
where σ is the sigmoid activation function.

Finally, the output $g^{h}$ and $g^{w}$ feature maps are expanded as the final weights of the CA module. The formula is as follows:(7)$$Y_{c}\left(i,j\right)=X_{c}\left(i,j\right)\times g_{c}^{h}\left(i\right)\times g_{c}^{w}\left(j\right)$$

From the implementation process of the CA module, it can be seen that CA module encodes the weights of the input feature map in two dimensions: height and width. Therefore, introducing the CA module into the segmentation network can effectively improve the ability of the model to convey high-dimensional semantic information.

#### 3.2.2. Cascade Feature Fusion

The multiscale feature layer is one of the important features of the backbone feature extraction network in DeeplabV3+. Among them, shallow features contain some local features that can obtain pixel-level localization accuracy, while deep features are global features that can obtain accurate contextual semantic information. Therefore, the efficient fusion of shallow and deep features can improve the feature representation ability of the network. In the DeepLabV3+ model, the decoding layer only splices and fuses the shallow and deep features of 1/4 size of the original image size, which leads to easy loss of the detailed information of the segmented object and insufficient segmentation accuracy. In order to obtain better segmentation performance, C-DeepLabV3+ employs a cascade feature fusion module to perform feature fusion on feature maps of different scales. The structure of the cascade feature fusion is shown in Figure 4.

The feature maps $F_{1}$ and $F_{2}$ are firstly used as inputs to the cascade feature fusion module, where $F_{1}$ and $F_{2}$ are feature maps of 1/8 size and 1/16 size of the original map, respectively. Then, the bilinear interpolation of $F_{2}$ is performed on 2-fold upsampling and *3 × 3* dilated convolution with a dilation rate of 2 to make the size and receptive field the same as $F_{1}$. At the same time, *1 × 1* convolution is performed on $F_{1}$ to match the number of feature channels of $F_{2}$. Next, a normalization operation is performed on the processed $F_{1}$ and $F_{2}$ using a Batch Normalization layer. Finally, the ReLU layer is used to sum the two features to obtain the fusion feature $F_{1}^{\prime }$. The fusion feature $F_{1}^{\prime }$ makes full use of the detail information of the input features and achieves the superposition of feature mapping for different scales. It avoids the problem of losing detail information in the road segmentation process and can effectively improve the precision and accuracy of road segmentation.

### 3.3. Vehicle Detection Based on UAV Aerial Images

UAV aerial images can have a high number of small targets and dense target density, thus making the YOLOv5 algorithm less able to cope with the challenges in these scenarios. This requires the detection algorithms to pay more attention to key regions in the image to improve the ability to extract features. Existing studies show that the attention mechanism is an effective means to improve the algorithm’s feature capturing ability by reducing the focus on other noncritical information so that the network can learn more useful information. In this paper, our proposed S-YOLOv5 algorithm adds a SimAM attention module to the neck network structure of YOLOv5, which is used to improve the detection accuracy of vehicle targets in complex environments under UAV perspective. The neck network structure diagram of S-YOLOv5 is shown in Figure 5.

Existing attention modules typically focus only on the channel domain or spatial domain, and these methods produce one-dimensional or two-dimensional weights that process neurons at each channel or spatial location equally, which may limit their ability to learn more discriminative cues. Complete three-dimensional weights are superior to traditional one- and two-dimensional attention, and features can be well refined using complete three-dimensional weights. Inspired by the attention mechanisms of the human brain, SimAM module is able to infer three-dimensional weights directly from the current neuron. It takes into account both spatial and channel dimensions in order to allow the network to learn more discriminative neurons. It is able to assign higher weights to important neurons without increasing the number of network parameters. The module schematic is shown in Figure 6.

SimAM module assigns higher weights to more important neurons by generating spatial inhibition on neighboring neurons, and it proposes the concept of an energy function to explore the importance of each neuron. After a series of simplifications, the expression for the minimum energy function is shown in Equation (Equation 8).
(8)$$e_{t}^{*}=\frac{4({\hat{\sigma }}^{2}+\lambda )}{{\left(t-\hat{\mu }\right)}^{2}+2{\hat{\sigma }}^{2}+2\lambda }$$
where *t* is the target neuron, λ is the hyperparameter, and $\hat{\mu }$ and ${\hat{\sigma }}^{2}$ denote the mean and variance on each channel, respectively, which are calculated as shown in Equations (Equation 9) and (Equation 10).
(9)$$\hat{\mu }=\frac{1}{M}\sum_{i=1}^{M}x_{i}$$
(10)$${\hat{\sigma }}^{2}=\frac{1}{M}\sum_{i=1}^{M}{\left(x_{i}-\hat{\mu }\right)}^{2}$$
where *x* is a neighboring neuron of *t*.

The lower the energy, the greater the difference between neuron *t* and the surrounding neurons, and the higher the importance. Therefore, the importance of a neuron can be obtained from $1/e_{t}^{*}$. The features are then augmented by Equation (Equation 11) to obtain the augmented feature tensor $\tilde{X}$.
(11)$$\tilde{X}=\mathrm{sigmoid}\;\left(\frac{1}{E}\right)⊙X$$
where *X* is the input feature tensor, *E* is the sum of $e_{t}^{*}$ over all channel and space dimensions, and ⊙ is the Hadamard product. The sigmoid function is used to limit excessive values of *E* without affecting the relative importance of each neuron.

The loss function of S-Yolov5 algorithm is shown in Equation (Equation 12).
(12)$$CIOU=IOU-\frac{\rho^{2}\left(b,b^{gt}\right)}{c^{2}}-\alpha v$$
where $\rho^{2}\left(b,b^{gt}\right)$ represent the Euclidean distances of the centroids of the predicted and real boxes, respectively. *c* represents the diagonal distance of the smallest closure region that can contain both the prediction box and the true box.

Compared to the YOLOv5 algorithm, the S-YOLOv5 algorithm can effectively capture the information of feature mapping in both global and local dimensions by embedding SimAM module to better detect and focus on important locations and features. It strengthens the focus on target objects and enhances the detection of small targets. At the same time, it is a good solution to the problem of high leakage detection rate and false detection rate for vehicles in dense scenes.

## 4. Experiment Results

### 4.1. Dataset

In this paper, we selected DroneVehicle [65], which is a dataset of UAV aerial vehicles released by Tianjin University. This dataset is a collection of aerial images of vehicle classes acquired from a high altitude viewpoint on a complex ground background, with a resolution of 840 × 712 for each image. Since not all the images in the dataset satisfy the experimental requirements, a total of 800 images containing clear road regions and vehicle targets on the road were selected from the dataset to form the experimental dataset. The experimental dataset was divided into training and test sets in the ratio of 7:3, where the training set had 560 images, and the test set had 240 images. Some sample images from the DroneVehicle dataset are shown in Figure 7.

Since the road segmentation experiments and the vehicle detection experiments have different requirements for the dataset, two different types of labeling processes are required for the dataset. For the road segmentation experiment, both the training process of the road segmentation model and the comparison of the test results require road labelled images, so labeling is a necessary step when conducting the experiment. The semantic segmentation annotation tool was utilized to annotate the road regions on the images and set the corresponding road labels, and finally, these images were converted to the VOC format for the training and testing of road segmentation experiments. For the vehicle target detection experiments, the vehicles were labelled on the images with their categories using the target detection annotation tool. Since the main vehicle categories in the 800 images were cars and trucks, they were divided into these two target categories and finally generated into the YOLO format.

### 4.2. Experimental Setting

The experiment was run on a Linux server with Ubuntu 18.04 (developed by Canonical in London, UK), a 12 vCPU Intel(R) Xeon(R) Platinum 8255C CPU @ 2.50 GHz, and RTX 3080 GPU with 10 GB of memory based on the Pytorch deep learning framework using Pytorch version 1.7.0 and CUDA version 11.0:In the road segmentation experiment, we trained and tested the C-DeepLabV3+ algorithm directly on the dataset. During the training process, freeze training was used to speed up the training efficiency, and the learning rate was set to $5\;\times \;10^{-4}$ for the freeze phase and $5\;\times \;10^{-5}$ for the unfreeze phase. In addition, the batch size was set to four, and 200 epochs were iterated.In the on-road vehicle detection experiment, the S-YOLOv5 was also trained and tested on the dataset. During training, the initial learning rate was set to $1\;\times \;10^{-3}$, the batch size was set to four, and 100 epochs were iterated.

### 4.3. C-DeepLabV3+ Experimental Analysis

The comparative experimental analysis semantic segmentation evaluation metrics used for road segmentation are detailed below.

PA (Pixel Accuracy): The ratio of the number of pixels predicted correctly for a category to the total number of pixels. It reflects the accuracy of the segmentation result for a particular category: the higher the metric, the better the algorithm, and the formula is Equation (Equation 13). *k* means the number of classes of labels, $p_{ii}$ means the amount of pixels accurately discriminated as class *i*, and $p_{ij}$ means the number of pixels belonging to class *i* but incorrectly predicted to be class *j*.
(13)$$\mathrm{PA}=\frac{\sum_{i=0}^{k}p_{ii}}{\sum_{i=0}^{k}\sum_{j=0}^{k}p_{jj}}$$

mPA (mean Pixel Accuracy): The ratio of the number of correctly classified pixels in each category label to the number of all pixels in the category label is calculated to obtain the PA of each category, and then the average index is obtained for all categories of PA, as shown in Equation (Equation 14). It is more reflective of the model’s ability to recognize various categories. k+1 means a category containing a background.
(14)$$\mathrm{mPA}=\frac{1}{k+1}\sum_{i=0}^{k}\frac{p_{ii}}{\sum_{j=0}^{k}p_{ij}}$$

mIoU (mean Intersection over Union): The ratio of the intersection and union between the prediction results of each category and the real label is calculated first, that is, the IoU, and then the IoU of all categories is summed and then averaged, as shown in Equation (Equation 15). Its value range is [0, 1], which is an important indicator to test the accuracy of segmenting objects in the dataset, and the larger the value of mIoU, the more accurate the segmentation result. $p_{ji}$ means the number of pixels belonging to class *j* but incorrectly predicted as class *i*.
(15)$$\mathrm{mIoU}=\frac{1}{k+1}\sum_{i=0}^{k}\frac{p_{ii}}{\sum_{j=0}^{k}p_{ij}+\sum_{j=0}^{k}p_{ji}-p_{ii}}$$

The C-DeepLabV3+ algorithm was trained on the working platform using the UAV aerial photography dataset, and the model’s superiority was measured using the PA, mPA, and mIoU. The road segmentation algorithm before and after improvement is shown in Table 1. The experimental results show that the proposed C-DeepLabV3+ algorithm improved the PA from 98.01% to 98.42%, which is an improvement of 0.41%, and there was an improvement of 0.2% in the mPA and 0.33% in the mIoU. This indicates that the segmentation accuracy of the improved algorithm improved to some extent and performed better in the test set.

The comparison of road segmentation results of UAV aerial images in some typical scenes is shown in Figure 8, Figure 9 and Figure 10. Among them, Figure 8 shows a comparison of the effectiveness of the road boundary segmentation. Figure 9 shows a comparison of the accuracy of the segmentation of road regions and nonroad regions, and Figure 10 shows a comparison of the effectiveness of the segmentation of shaded regions of the road.

As can be seen from Figure 8, the DeepLabV3+ algorithm is more fuzzy in the segmentation of the road boundary, and the segmentation at the boundary has the problem of unevenness. Our proposed C-DeepLabV3+ algorithm has a smoother and clearer segmentation profile at the road boundary, and less road information is lost at the boundary. Therefore, the C-DeepLabV3+ algorithm is better in segmentation at the road boundary.

It can be seen from Figure 9 that the DeepLabV3+ algorithm had poor recognition results, with deviations in the segmentation of road and nonroad regions and low road recognition accuracy. In contrast, the C-DeepLabV3+ algorithm was able to better discriminate road features, and the recognition of road and nonroad regions is significantly accurate. Meanwhile, the segmentation fineness and continuity were improved, and better road segmentation results were obtained. Therefore, the C-DeepLabV3+ algorithm has higher road segmentation accuracy and better performance.

As can be seen in Figure 10, the other algorithms incorrectly recognized the shadow regions formed on the road by tree occlusion as nonroad regions. However, the C-DeepLabV3+ algorithm could achieve more accurate detection, which well solves the problem of dealing with the effects of occlusion and shadows on road segmentation. Also, the the C-DeepLabV3+ algorithm works best when it performs a special type of road segmentation like intersections. Therefore, the C-DeepLabV3+ algorithm can effectively solve the problem of recognizing shadows on road regions.

In summary, the road segmentation algorithm C-DeepLabV3+ proposed in this paper has been effectively improved in the face of poor road edge segmentation, the inaccurate identification of road and nonroad regions, and the wrong identification of shadow regions on the road. It is more suitable for the extraction of UAV aerial images and has achieved better results and performance.

### 4.4. S-YOLOv5 Experimental Analysis

The evaluation indicators of the comparison test for vehicle detection are as follows.

Precision is the ratio of the number of true positive instances predicted to the number of all instances predicted to be positive: the higher the indicator, the higher the accuracy of the algorithm, and the formula is Equation (Equation 16). TP means the true positive rate, which is the number of instances where the predicted result is positive, and the true result is also positive. FP means the false positive rate, which is the number of instances where the predicted result is positive, but the true result is negative.
(16)$$\mathrm{Precision}=\frac{\mathrm{TP}}{\mathrm{TP}+\mathrm{FP}}$$

Recall is the ratio of predicted true positive cases to the total number of actual positive cases: the higher the indicator, the better the algorithm, as shown in Equation (Equation 17). FN means the false negative rate, which is the number of instances where the predicted result is negative, but the true result is positive.
(17)$$\mathrm{Recall}=\frac{\mathrm{TP}}{\mathrm{TP}+\mathrm{FN}}$$

The Average Precision (AP) considers both Precision and Recall, with Recall as the horizontal axis and Precision as the vertical axis to form the PR curve. The area covered by the curve is the AP, which measures the quality of the model in a single category. The formula is as follows:(18)$$\mathrm{AP}=\int_{0}^{1}P\left(R\right)\mathrm{dR}$$

The mean Average Precision (mAP) refers to the average of the average precision across all categories and can be used to compare the performance of different algorithms side by side. Generally, the threshold is set to 0.5, that is, the prediction box with an IoU greater than 0.5 is valid and denoted by mAP@0.5, as shown in Equation (Equation 19).
(19)$$\mathrm{mAP}=\frac{1}{N}\sum_{i=1}^{N}{\mathrm{AP}}_{i}$$

From Table 2, it can be seen that our algorithm S-YOLOv5 improved in all evaluation metrics, with 1.0% in Precision, 1.15% in Recall, and 0.45% in mAP. The experimental results show that the proposed S-YOLOv5 algorithm has better performance.

A comparison of the vehicle detection results of the UAV aerial images in a plurality of different scenarios is illustrated in Figure 11, Figure 12 and Figure 13.

The comparative results of the detection of small target vehicles under UAV aerial images are shown in Figure 11.

As can be seen from Figure 11, our method could recognize more cars with smaller sizes than others, which shows that the S-YOLOv5 algorithm is able to recognize more vehicles, and it is more effective for the detection of small targets in the UAV aerial images. The S-YOLOv5 algorithm reduced the leakage rate of the small targets, and it could solve the problem of the difficulty of detecting small targets in the UAV aerial images relatively well.

Figure 12 shows the vehicle detection results of different methods for congestion scenarios in UAV aerial images.

As can be seen in Figure 12, the SSD and YOLOv5 algorithms missed vehicle targets, while the S-YOLOv5 algorithm was able to correctly identify vehicle targets and nonvehicle targets. In addition, the Faster R-CNN and YOLOv5 algorithms incorrectly recognized road signs as vehicle targets, while the S-YOLOv5 algorithm achieved the correct categorization of detected vehicle targets. As a result, S-YOLOv5 can effectively reduce the rate of misdetection and improve the accuracy of vehicle detection.

The comparative results of vehicle detection under complex background conditions in the UAV aerial images are shown in Figure 13.

As can be seen in Figure 13, the SSD and YOLOv5 algorithms failed to detect some cars with distinctive features, and vehicle types were not accurately distinguished. The Faster R-CNN detection was even worse, thereby not only failing to detect valid vehicle targets but also misclassifying targets that are clearly not cars. In contrast, the S-YOLOv5 algorithm was able to detect these car targets correctly. In addition, the SSD and YOLOv5 did not detect the truck, and the S-YOLOv5 successfully detected the truck, which indicates that it has better detection performance for both cars and trucks and effectively reduces the miss detection rate.

In summary, our proposed S-YOLOv5 algorithm significantly improved the recognition accuracy compared to other algorithms, and it has better detection of small targets. At the same time, the leakage rate and false detection rate were also significantly reduced, and it is more suitable for vehicle detection under the UAV perspective.

### 4.5. Segmentation–Detection Framework Analysis

Based on the segmentation–detection framework, the road segmentation algorithm and the vehicle detection algorithm were jointly experimented, and the pipeline of the two tasks has been realized. The final results of road segmentation and vehicle detection in different viewpoints and scenes can be obtained as shown in Figure 14 and Figure 15.

As can be seen from Figure 14, the C-DeepLabV3+ algorithm was able to extract the road regions accurately, and the segmentation of the road boundaries are clear in the ordinary scene. S-YOLOv5 could detect the vehicle targets on the road with high detection accuracy.

As can be seen from Figure 15, C-DeepLabV3+ could accurately recognize road regions and nonroad regions in dense scenes. The segmentation of the road boundaries are clear. This algorithm effectively solves the occlusion problem, which provides a good foundation for subsequent traffic target detection on the road. For the problem of dense vehicle targets and many small targets, S-YOLOv5 had a high detection accuracy, which could correctly detect all the vehicle targets on the road and finally obtained accurate experimental results.

In summary, our proposed road segmentation algorithm C-DeepLabV3+ can accurately extract road regions under different viewpoints, and our proposed vehicle detection algorithm S-YOLOv5 can accurately detect vehicle targets under various scenarios. The experimental results are satisfactory, and the serial pipeline of road segmentation and vehicle detection on the road can be realized with high quality.

## 5. Conclusions

In this paper, we proposed a novel and efficient network framework for simultaneous road segmentation and vehicle detection to process UAV aerial traffic images, which can facilitate the construction of smart transportation. The proposed C-DeepLabv3+ road segmentation algorithm introduces the CA module to enhance the focus on road information, and it adds a CFF module to prevent the loss of road detail information. The proposed S-YOLOv5 vehicle detection algorithm introduces the parameter-free attention module SimAM for enhancing the vehicle’s ability to detect small targets and improve detection accuracy. Our proposed network framework combines the two improved algorithms with higher road segmentation accuracy and more accurate vehicle detection, and it demonstrates superior performance and results on the constructed DroneVehicle dataset. In the future, the framework model can be considered to be lightweighted for piggybacking on edge devices for real-time segmentation and detection.

## Figures and Tables

**Figure 1 sensors-24-03606-f001:**
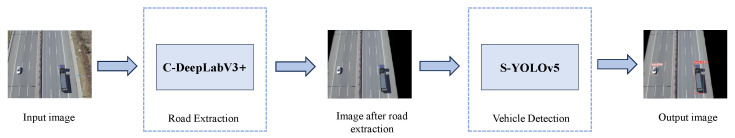
Road segmentation–vehicle detection framework implementation diagram.

**Figure 2 sensors-24-03606-f002:**
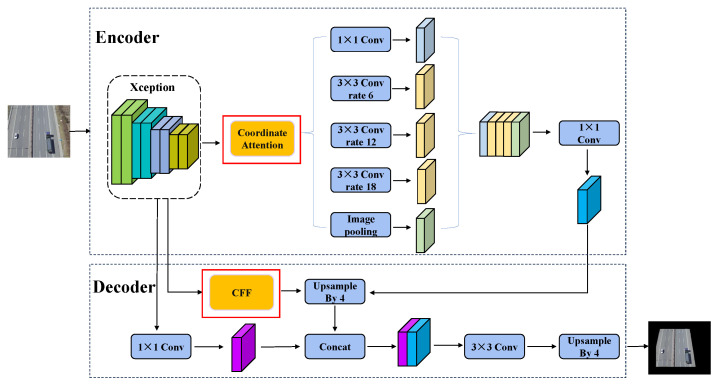
Architecture of C-DeepLabV3+.

**Figure 3 sensors-24-03606-f003:**
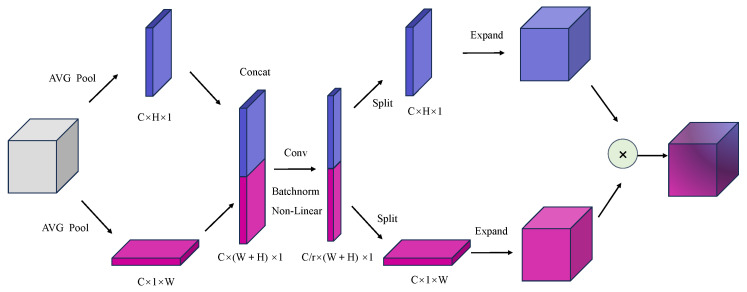
Coordinate attention module schematic [62].

**Figure 4 sensors-24-03606-f004:**
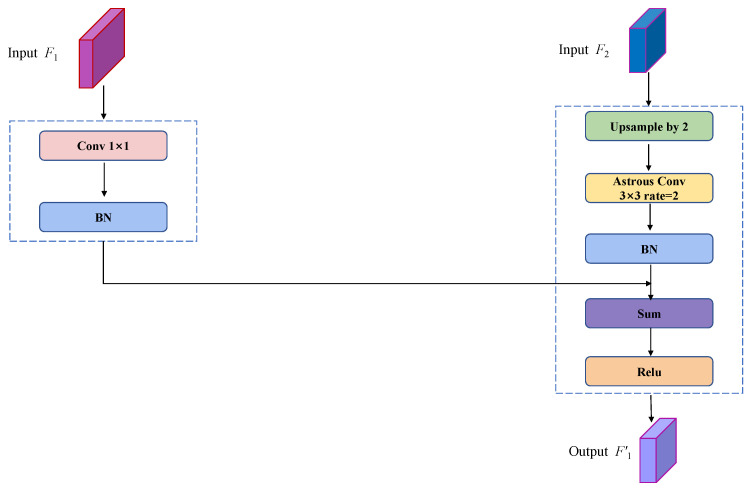
Cascade feature fusion structure diagram [63].

**Figure 5 sensors-24-03606-f005:**
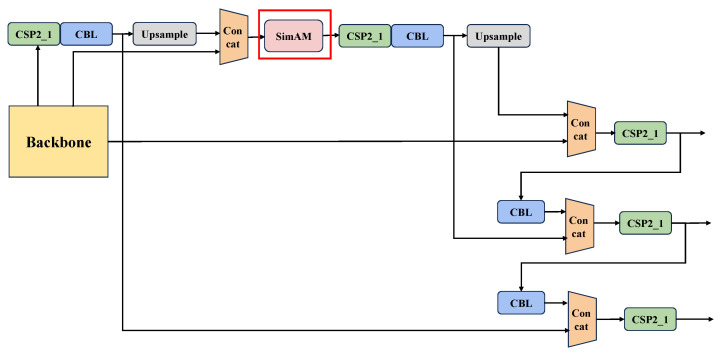
Neck network structure diagram of S-YOLOv5.

**Figure 6 sensors-24-03606-f006:**
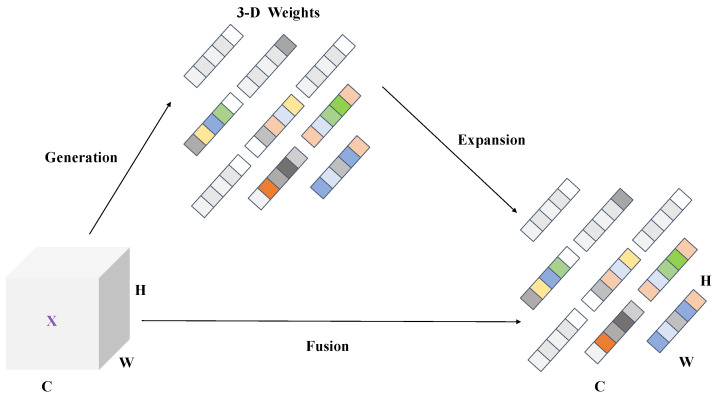
SimAM module schematic [64].

**Figure 7 sensors-24-03606-f007:**
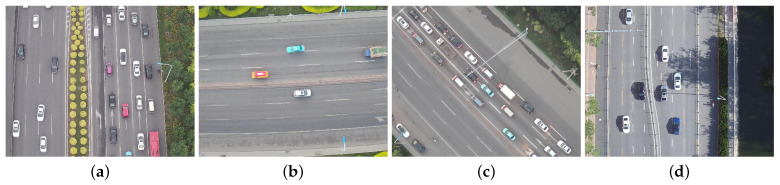
(**a**–**d**) Sample images from the DroneVehicle dataset.

**Figure 8 sensors-24-03606-f008:**
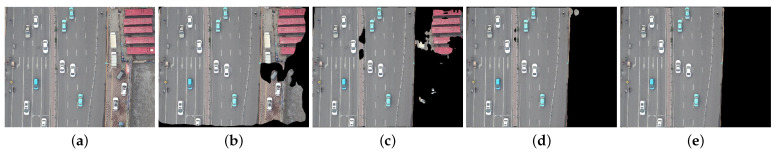
Comparison of road boundary segmentation results on different methods. (**a**) Original image; (**b**) PSPNet; (**c**) U-Net; (**d**) DeepLabV3+; (**e**) C-DeepLabV3+.

**Figure 9 sensors-24-03606-f009:**
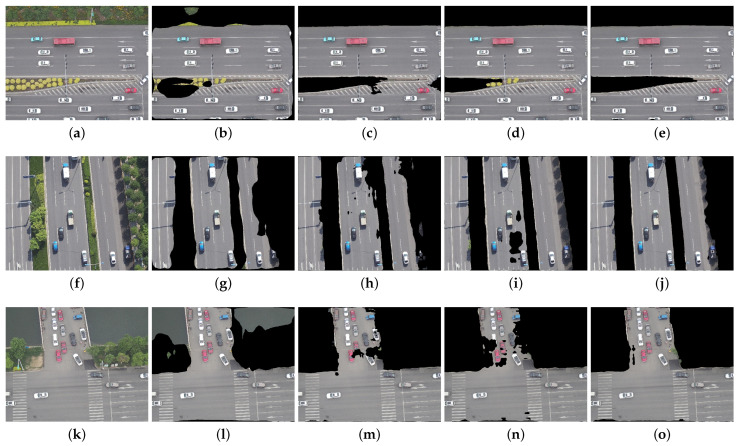
Comparison of segmentation accuracy results for road and nonroad regions using different methods. (**a**,**f**,**k**) Original images; (**b**,**g**,**l**) PSPNet; (**c**,**h**,**m**) U-Net; (**d**,**i**,**n**) DeepLabV3+; (**e**,**j**,**o**) C-DeepLabV3+.

**Figure 10 sensors-24-03606-f010:**
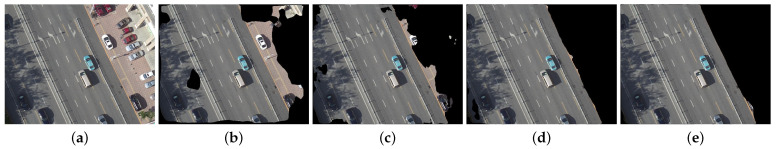
Comparison of segmentation results for road shadow regions on different methods. (**a**) Original image; (**b**) PSPNet; (**c**) U-Net; (**d**) DeepLabV3+; (**e**) C-DeepLabV3+.

**Figure 11 sensors-24-03606-f011:**
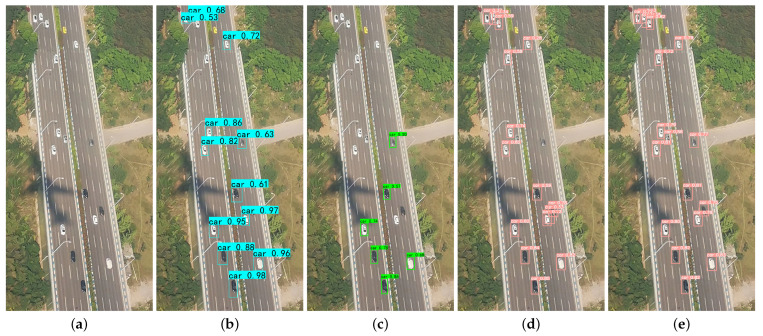
Comparison results of detection of small target vehicles using different methods. (**a**) Original image; (**b**) Faster R-CNN; (**c**) SSD; (**d**) YOLOv5; (**e**) S-YOLOv5.

**Figure 12 sensors-24-03606-f012:**
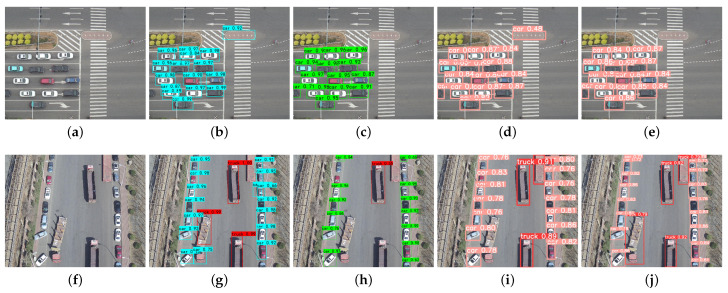
Comparison of vehicle detection results for congestion scenarios using different methods. (**a**,**f**) Original image; (**b**,**g**) Faster R-CNN; (**c**,**h**) SSD; (**d**,**i**) YOLOv5; (**e**,**j**) S-YOLOv5.

**Figure 13 sensors-24-03606-f013:**
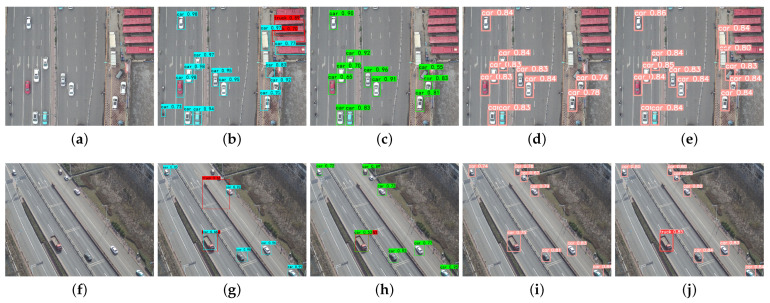
Comparison of vehicle detection results under complex background conditions. (**a**,**f**) Original image; (**b**,**g**) Faster R-CNN; (**c**,**h**) SSD; (**d**,**i**) YOLOv5; (**e**,**j**) S-YOLOv5.

**Figure 14 sensors-24-03606-f014:**
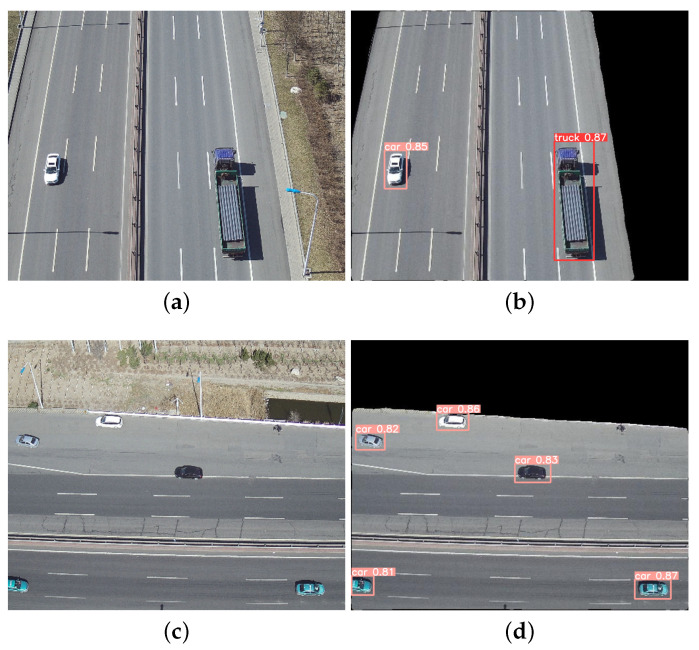
Segmentation–detection results in common scenarios of UAV aerial photography in the same framework. (**a**,**c**) Original images. (**b**,**d**) Result images.

**Figure 15 sensors-24-03606-f015:**
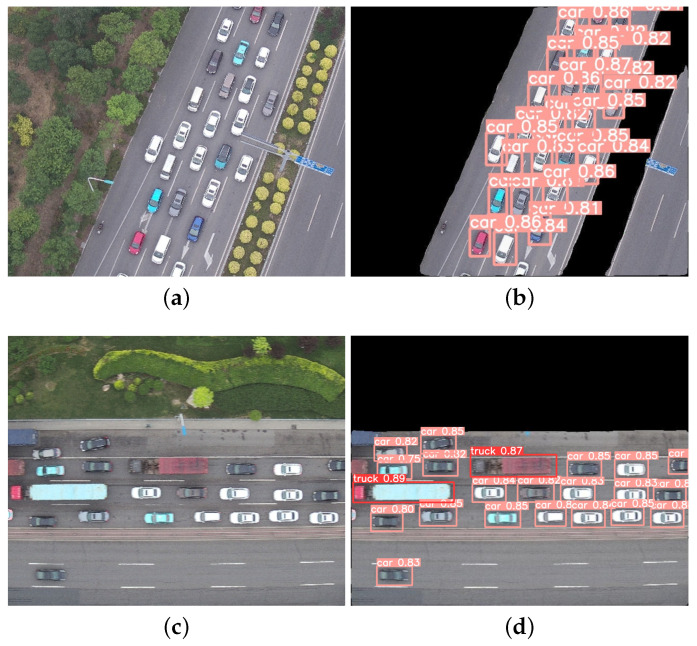
Segmentation–detection results in intense scenarios of UAV aerial photography in the same framework. (**a**,**c**) Original images. (**b**,**d**) Result images.

**Table 1 sensors-24-03606-t001:** Comparison of road segmentation metrics on different methods.

Methods	PA	mPA	mIoU
PSPNet [66]	96.20%	95.61%	90.98%
U-Net [43]	96.35%	98.54%	97.09%
DeepLabV3+	98.01%	98.55%	97.20%
C-DeepLabV3+	98.42%	98.75%	97.53%

**Table 2 sensors-24-03606-t002:** Comparison of vehicle detection metrics using different methods.

Methods	Precision			Recall			AP		mAP@0.5
	Car	Truck	All	Car	Truck	All	Car	Truck	
Faster R-CNN [9]	53.48%	61.89%	57.79%	93.14%	89.80%	91.47%	75.70%	86.44%	81.07%
SSD [55]	92.79%	89.51%	91.15%	90.15%	56.86%	73.51%	95.82%	80.61%	88.21%
YOLOv5	96.80%	88.30%	92.50%	97.30%	95.20%	96.30%	97.80%	96.10%	96.95%
S-YOLOv5	97.10%	90.00%	93.50%	97.70%	97.20%	97.45%	98.10%	96.70%	97.40%

## Data Availability

The datasets are available on Github at https://github.com/VisDrone/DroneVehicle; accessed on 20 August 2022.

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
