# Peer review of "A Novel Network Framework on Simultaneous Road Segmentation and Vehicle Detection for UAV Aerial Traffic Images"

_sensors, 2024, doi:10.3390/s24113606_

Round 1
Reviewer 1 Report
Comments and Suggestions for Authors
The paper proposes a lane segmentation algorithm C-DeepLabV3+ and vehicle detection algorithm S-YOLOv5 for UAV aerial images, and proposes an algorithmic framework to complete lane segmentation and vehicle detection in the same frame. The authors illustrate the superiority of the proposed algorithms through experiments. However, there are several points can be improved in the manuscript:
1) The authors propose a road segmentation algorithm in the title, and the diversity of road information that including different road types, shapes, locations, and sign lines, etc., but the road segmentation algorithm proposed in the paper only recognizes the road area and does not segment each lane accordingly, and it is suggested that the authors can increase the segmentation of each lane. What’s more, from the application point of view, the road segmentation result only includes drivable areas, which may be difficult to meet the map requirements of downstream applications of the data, for example, the road segmentation results of the highD and AD4CHE datasets are in units of each lane, and it is possible to get the lane id of where each vehicle is located. Therefore, the output of the map at the lane level will increase the value of the application of the algorithm proposed in the paper.
2) The dataset used for vehicle detection in the paper is small, containing only 800 images, of which only 560 are used for training. Considering that the training of deep learning models requires a large amount of data support, this dataset size may lead to insufficient generalization of the model and affect the practical application of the algorithm. It is recommended that the authors expand the dataset size and conduct more adequate model validation and evaluation in subsequent studies.
3) Road types applicable to the algorithm can be supplemented/defined: whether it applies to special road types (e.g., intersections, traffic circles, parking lots, etc.).
4) Subsequent data processing process to be added/specified: Usually, the original data of the downstream applications are a time series of multi-frame images. Given the instability of the UAV's shooting and the impact of the vehicle position changing in each frame, can the algorithm in this paper can ensure the stability of the results of the drivable area? At the same time, how to deal with the tasks of target tracking and coordinate alignment between map and vehicles, which are most likely to be needed in real applications, needs to be explained/improved. It is suggested to add/clarify whether the algorithm has a real adaptation problem in practical applications.
Author Response
Dear reviewer:
We would like to thank you for the valuable comments on our manuscript entitled " A Novel Network Framework on Simultaneous Road Segmentation and Vehicle Detection for UAV Aerial Traffic Images". The comments were productive and helped us improve the quality of the manuscript. Based on your comments, we carefully revised the manuscript. The point-to-point response is as follows:
Point 1: The authors propose a road segmentation algorithm in the title, and the diversity of road information that including different road types, shapes, locations, and sign lines, etc., but the road segmentation algorithm proposed in the paper only recognizes the road area and does not segment each lane accordingly, and it is suggested that the authors can increase the segmentation of each lane. What’s more, from the application point of view, the road segmentation result only includes drivable areas, which may be difficult to meet the map requirements of downstream applications of the data, for example, the road segmentation results of the highD and AD4CHE datasets are in units of each lane, and it is possible to get the lane id of where each vehicle is located. Therefore, the output of the map at the lane level will increase the value of the application of the algorithm proposed in the paper.
Response 1:
Thank you for your insightful comments. This paper is an algorithmic framework for both lane segmentation and vehicle detection in the same frame, for the lane segmentation experiment identifies the drivable area of the road for subsequent vehicle detection on the drivable road. It can be applied to monitoring the road, traffic flow detection, traffic management, and can be integrated with the intelligent transportation system to support the related work of the transportation department. The segmentation you suggested for each lane is a possible direction for future research and application because of the non-existence of lane-related information in the dataset, and your professional advice is greatly appreciated.
Point 2: The dataset used for vehicle detection in the paper is small, containing only 800 images, of which only 560 are used for training. Considering that the training of deep learning models requires a large amount of data support, this dataset size may lead to insufficient generalization of the model and affect the practical application of the algorithm. It is recommended that the authors expand the dataset size and conduct more adequate model validation and evaluation in subsequent studies.
Response 2:
Thanks for your professional advice. The vehicle detection dataset we used contains UAV vehicle traffic images captured at various angles, including ordinary scenes, dense scenes, and small targets, covering enough scenes for the dataset to be rich enough. The experimental evaluation of the model shows high detection accuracy and low leakage and false detection rates for different kinds of images to be detected.
Point 3: Road types applicable to the algorithm can be supplemented/defined: whether it applies to special road types (e.g., intersections, traffic circles, parking lots, etc.).
Response 3:
Thank you for your insightful comment. The explanation is as follows:
The algorithm proposed in this paper mainly detects drivable areas on the road and moving vehicles on the road, it is not applicable to parking lots. Traffic circles are not an applicable target due to the presence of interference from multiple roads. The C-DeepLabV3+ algorithm in this paper can be used for intersections. It has been supplemented and highlighted in the manuscript and corresponding experimental results have been added in Figure 9. The details are as follows:
This road segmentation algorithm can handle ordinary roads and intersections.
Point 4: Subsequent data processing process to be added/specified: Usually, the original data of the downstream applications are a time series of multi-frame images. Given the instability of the UAV's shooting and the impact of the vehicle position changing in each frame, can the algorithm in this paper can ensure the stability of the results of the drivable area? At the same time, how to deal with the tasks of target tracking and coordinate alignment between map and vehicles, which are most likely to be needed in real applications, needs to be explained/improved. It is suggested to add/clarify whether the algorithm has a real adaptation problem in practical applications.
Response 4:
Thank for your professional advice.. The explanation is as follows:
Our algorithm has demonstrated superior performance in processing single-frame images, and time series of multi-frame images consisting of a single frame. After extensive experiments by researchers and the industry, the semantic segmentation algorithm DeepLabV3+ itself employs the techniques of null convolution and multi-scale feature fusion, and shows superior segmentation results for dynamic scenes. Our proposed C-DeepLabV3+ algorithm introduces the CA attention module, which can more accurately localize the segmented target information and is better suited for dynamic scenes even with the instability of UAV shooting. The target detection algorithm Yolov5 itself shows good stability in processing time series composed of multi-frame images, and the SimAM module added to our proposed vehicle detection algorithm S-Yolov5 is more sensitive to the capture of vehicle targets, so that even if the vehicle position changes in each frame, it does not affect the stability of the detection results. The stability of the results can be further improved in the future by other techniques such as motion compensation, smoothing, and inter-frame information fusion.
For the problem of dealing with the target tracking and coordinate alignment tasks between the map and the vehicle, our proposed framework can be integrated into a SLAM system for dynamic multi-target tracking to achieve precise positioning on the map, which is a particularly worthwhile application direction for future research.
Our proposed C-DeepLabV3+ and S-Yolov5 algorithms have certain environmental adaptability and demonstrate excellent results for different image scenes. At the same time, our algorithms can be migrated to the field of automatic driving, agriculture and other applications, which need to be adjusted according to the specific application scenarios. Thanks again for your professional advice.
Thanks again to the valuable comments made by the review expert. Although we strive to create and try to express our academic achievements, there are still many shortcomings. The suggestions you have given us are very pertinent, which also makes our papers go up the ladder. Thank you!
Finally, take this opportunity to wish you good health and work well!

Reviewer 2 Report
Comments and Suggestions for Authors
Decision: Major Revision
(Language issue) Please check the manuscript again since there are serious grammatical errors throughout the manuscript and dispersed typos. Authors are advised to go through the manuscript.
From the abstract, it is not clear actual which problem is targeted in what context. The clarity is not there.
The author should clearly explain the performance of the proposed method in the abstract section.
Please only use one single abbreviation convention, for example Deep Learning (DL) or deep learning (DL) please check for inconsistencies.
The authors are recommended to include the following articles which closely align with your research to solidify the introduction and motivation section.
o https://doi.org/10.32604/csse.2023.037992
o https://doi.org/10.1016/j.eswa.2024.123935
Please explain more about the coordinate attention, what is the actual modification in the attention module.
Figure 7, 8 and 9 I am curious to know how the proposed model works here, the figure is about the qualitative results, but the details are not clear, please explain.
The conclusion lakes clarity, please re-write this section for ease of the reader.
Figure 6 the authors are depicting 3D weights expansion, but I don't see any difference visually in the architecture, please explain how it works.
Some sample images from the employed dataset should be added for readers' interest.
As a reader I don't understand the segmentation of just the road boundary, how it can be useful for the model? As the model is only looking for the annotated items on the road, i.e., vehicles, so how are these relevant, please explain the whole work properly.
Please add an equation for the loss of the model in a similar manner like equation 15, 16,17 etc.
I am curious to know the performance of other variants of YOLO with different backbones and feature extractors for comparison in Table 2.
Author Response
Dear reviewer:
We would like to thank you for the valuable comments on our manuscript entitled " A Novel Network Framework on Simultaneous Road Segmentation and Vehicle Detection for UAV Aerial Traffic Images". The comments were productive and helped us improve the quality of the manuscript. Based on your comments, we carefully revised the manuscript. The point-to-point response is as follows:
Point 1: Please check the manuscript again since there are serious grammatical errors throughout the manuscript and dispersed typos. Authors are advised to go through the manuscript.
Response 1:
Thank you very much for good advice. In order to enhance the rigor and standardization of this paper, we carefully checked the entire paper and did our best to correct grammatical errors and scattered typos.
Point 2: From the abstract, it is not clear actual which problem is targeted in what context. The clarity is not there.
Response 2:
Thank you for your professional advice. We have revised the abstract to provide a detailed and clear description of the research context, and the problem addressed in this paper. The details are as follows:
Unmanned Aerial Vehicle (UAV) aerial sensors are an important means of collecting ground image data. Through road segmentation and vehicle detection of drivable areas in UAV aerial images, they can be applied to monitoring roads, traffic flow detection, traffic management, etc., and can be integrated with intelligent transportation systems to support the related work of transportation departments. Existing algorithms only realize a single task, while intelligent transportation requires the simultaneous processing of multiple tasks, which cannot meet the complex practical needs.
Please refer to the red text in the abstract of this paper.
Point 3: The author should clearly explain the performance of the proposed method in the abstract section.
Response 3:
Thanks for your professional comment. Based on your suggestion, we explain in detail the performance advantages of the proposed method in this paper in the abstract section. The details are as follows:
The experimental results show that on the constructed ViDroneVehicle dataset, the C-DeepLabV3+ algorithm has an mPA value of 98.75% and an mIoU value of 97.53%, which can better segment the road area and solve the problem of occlusion.The mAP value of the S-YOLOv5 algorithm has an mAP value of 97.40%, which is more than YOLOv5's 96.95%, which effectively reduces the vehicle omission and false detection rates. By comparison, the results of both algorithms are superior to multiple state-of-the-art methods. The overall framework proposed in this paper has superior performance and is capable of realizing high-quality and high-precision road segmentation and vehicle detection from UAV aerial images.
Please refer to the red text in the abstract of this paper.
Point 4: Please only use one single abbreviation convention, for example Deep Learning (DL) or deep learning (DL) please check for inconsistencies.
Response 4:
Thanks for your professional comment. We carefully checked the entire paper and standardized all the abbreviations. The deep learning abbreviations you pointed out have all been changed to deep learning (DL) to ensure consistency.
Point 5: The authors are recommended to include the following articles which closely align with your research to solidify the introduction and motivation section.
https://doi.org/10.32604/csse.2023.037992
https://doi.org/10.1016/j.eswa.2024.123935
Response 5:
Thank you for your professional advice. We have included the references you mentioned in the introduction section, as shown in references 5 and 6. This makes the motivation of this paper clearer and deeper. Please refer to the highlighted text in the introduction section of the paper.
Point 6: Please explain more about the coordinate attention, what is the actual modification in the attention module.
Response 6:
Thanks for your professional comment. The explanation is as follows:
Our proposed C-DeepLabV3+ road segmentation algorithm introduces a CA attention module in the encoder. It is capable of acquiring target-aware location-sensitive data and helps to improve the accuracy of segmentation target location localization and recognition of targets of interest. The detailed principles and roles of the CA module are explained in the Methods section. The experiments show that the road segmentation precision and accuracy increase after adding the CA attention module, proving that the improvement is effective.
Point 7: Figure 7, 8 and 9 I am curious to know how the proposed model works here, the figure is about the qualitative results, but the details are not clear, please explain.
Response 7:
Thank you very much for good advice. The explanation is as follows:
The coding layer first adopts Xception as the model backbone network to extract road features from the input UAV aerial images, and the backbone network inputs the extracted semantic features of different sizes into the CA attention module, which can obtain more accurate segmentation target location information, make the segmentation target edges more continuous, and improve the segmentation accuracy effectively. In the decoder, the shallow and deep feature maps of different sizes extracted by the backbone network are feature fused by our proposed CFF module, which improves the network's characterization ability and captures more accurate location information.
For the output results, the output image after segmentation should be a 24-bit color segmentation map. In order to clearly show the road regions in the image, we have blended the segmentation map with the original image, resulting in an image with black areas representing non-road regions and retained as road regions. These blended images will be used as input data for subsequent on-road vehicle detection targets in the framework.
Point 8: The conclusion lakes clarity, please re-write this section for ease of the reader.
Response 8:
Thank you for your professional advice. We have re-write the conclusion subsection to more clearly summarize the improvements, strengths, and possible future research directions of the methodology of this paper. It has been revised and marked yellow in the manuscript.
Point 9: Figure 6 the authors are depicting 3D weights expansion, but I don't see any difference visually in the architecture, please explain how it works.
Response 9:
Thanks reviewer for good question. In the description of the SimAM attention module(Subsection 3.3), we add the functional advantages of using 3D weights over 1D or 2D weights. The details are as follows:
Existing attention modules typically focus only on the channel domain or spatial domain, and these methods produce one- or two-dimensional weights and process neurons at each channel or spatial location equally, which may limit their ability to learn more discriminative cues. Complete three-dimensional weights are superior to traditional one- and two-dimensional attention, and features can be well refined using complete three-dimensional weights.SimAM moudle, inspired by the attentional mechanisms of the human brain, is able to infer three-dimensional weights directly from the current neuron, i.e., it takes into account the consideration of both the spatial and the channel dimensions in order to allow the network to learn more discriminative neurons. It is able to assign higher weights to important neurons without increasing the number of network parameters.
Please refer to the highlighted words in paper.
Point 10: Some sample images from the employed dataset should be added for readers' interest.
Response 10:
Thanks reviewer for good advice. We have added some sample images from the employed dataset in subsection 3.3 Dataset, as shown in Fig. 7. This increases the professionalism of the article and captures the reader's interest more.
Point 11: As a reader I don't understand the segmentation of just the road boundary, how it can be useful for the model? As the model is only looking for the annotated items on the road, i.e., vehicles, so how are these relevant, please explain the whole work properly.
Response 11:
Thanks reviewer for good advice. The explanation is as follows:
This paper is in the algorithmic framework of simultaneous road segmentation and vehicle detection on UAV aerial traffic images, which needs to use the road segmentation model to identify the drivable road area in the picture, and then according to the image after the road extraction has been performed, the detection of vehicle targets on drivable roads, which can be applied to monitoring roads, traffic flow detection, traffic management, etc., and is able to be integrated with the intelligent transportation system. The accuracy of road segmentation will directly affect the accuracy of vehicle detection, otherwise targets on non-road areas will be detected.
Point 12: Please add an equation for the loss of the model in a similar manner like equation 15, 16,17 etc.
Response 12:
Thanks for your professional advice. We added loss functions for algorithms C-DeepLabV3+ and S-YOLOv5 as shown in Eqs. (1) and (12), respectively. Please refer to the highlighted words in paper.
Point 13: I am curious to know the performance of other variants of YOLO with different backbones and feature extractors for comparison in Table 2.
Response 13:
Thanks for your professional advice. In this paper, the most typical representative of YOLO, YOLOv5, which is the most widely used and stable YOLO series algorithm, is selected to innovate it. Due to the simplicity of its network structure, it also facilitates subsequent improvement and research.YOLOv5 is also compared with other models, such as Faster RCNN and SSD, to highlight the excellence of the innovated method.
Thanks again to the valuable comments made by the review expert. Although we strive to create and try to express our academic achievements, there are still many shortcomings. The suggestions you have given us are very pertinent, which also makes our papers go up the ladder. Thank you!
Finally, take this opportunity to wish you good health and work well!

Round 2
Reviewer 1 Report
Comments and Suggestions for Authors
The revised manuscript has solved all the comments I mentioned before, it is suitable to be published in current version.
Reviewer 2 Report
Comments and Suggestions for Authors
The authors have addressed all comments, therefore I recommend to publish this article in present form.